# Socioeconomic Disparities and Risk of Papillary Thyroid Cancer Associated with Environmental Exposure to Per- and Polyfluoroalkyl Substances (PFAS) in Florida

**DOI:** 10.3390/ijerph22081290

**Published:** 2025-08-18

**Authors:** Laura E. Wild, Nicholas DiStefano, Garrett Forman, Bianca I. Arocha, Ming S. Lee, Peter A. Borowsky, Elizabeth Franzmann, Natasha Solle, Alberto J. Caban-Martinez, Erin Kobetz

**Affiliations:** 1Department of Otolaryngology, Miller School of Medicine, University of Miami, Miami, FL 33136, USA; 2Department of Public Health Sciences, Miller School of Medicine, University of Miami, Miami, FL 33136, USA; 3Sylvester Comprehensive Cancer Center, Miller School of Medicine, University of Miami, Miami, FL 33136, USA

**Keywords:** per- and polyfluoroalkyl substances, papillary thyroid cancer, environmental exposure, environmental justice, socioeconomic disparities

## Abstract

The existing literature suggests that exposure to Per- and Polyfluoroalkyl Substances (PFAS) can increase Papillary Thyroid Cancer (PTC) risk by interfering with thyroid hormone signaling, leading to hormonal imbalances that promote carcinogenesis. In addition, significant disparities exist in environmental exposure. However, ecological evidence of these associations has not been established within a statewide database of cancer outcomes. Therefore, this study investigated the relationship between socioeconomic conditions, environmental PFAS exposure, and PTC incidence in Florida using the state’s cancer registry. Data on facilities potentially releasing PFAS and ZIP codes with known PFAS drinking water contamination were retrieved from the EPA’s PFAS Analytic Tool. Proximity to PFAS sites and age-adjusted incidence by patient race/ethnicity were calculated by census tract. Lower socioeconomic status was associated with greater exposure to environmental PFAS. Census tracts with closer proximity to PFAS sites were more likely to have public water systems with PFAS contamination. Lastly, residential proximity to PFAS sites was positively associated with age-adjusted PTC incidence in Non-Hispanic Whites and Hispanics. These results demonstrate disparities in environmental exposure and suggest that exposure to PFAS may be an important factor for PTC risk at the population level and should be considered in the development of public health policies.

## 1. Introduction

Thyroid cancer is the most common endocrine cancer in the United States, with 44,000 new cases estimated for 2025 [1]. Approximately 80–85% of all thyroid cancer cases are Papillary Thyroid Cancer (PTC). Although PTC has an excellent prognosis, with a 99% 5-year relative survival rate for localized/regional cases and 74% for metastasized cases, disparities in survival persist [1]. For example, in the United States, racial and ethnic minority and lower socioeconomic status patients are more likely to experience delays in diagnosis and often present with advanced-stage disease, resulting in higher mortality rates compared to their White and more advantaged counterparts [2]. Moreover, because most cases of PTC are highly curable with thyroid lobectomy or total thyroidectomy, which necessitates life-long thyroid hormone replacement therapy, overall healthcare costs associated with all thyroid cancer are substantial. Overall costs to the United States health system associated with the diagnosis, treatment, and maintenance of thyroid cancers is predicted to reach USD 3.5 billion dollars by 2030 [3]. Thus, investigations on preventable risk factors are important and urgent for reducing the incidence, disparities, and overall healthcare costs associated with PTC.

Currently recognized PTC risk factors include prior radiation exposure, iodine deficiency, female sex, obesity, and certain familial syndromes, such as Gardner’s syndrome, Cowden disease, Carney complex, and Werner syndrome [4,5,6,7,8]. Emerging research also suggests that exposure to per- and polyfluoroalkyl substances (PFAS) may contribute to endocrine-related cancers, including PTC [9]. Chemicals containing PFAS are widely used in industrial applications and consumer products and are often labeled “forever chemicals” due to the strong carbon-fluorine bonds in their molecular structures [10]. It has been hypothesized that PFAS increases PTC risk by interfering with thyroid hormone signaling, potentially leading to hormonal imbalances that promote carcinogenesis. It was discovered that elevated blood PFAS levels are associated with an increased risk of hypothyroidism, further supporting the chemicals’ endocrine disrupting potential [11,12,13,14,15,16]. A 2023 study found that higher levels of perfluorooctanesulfonic acid in plasma, a congener of PFAS, were associated with a 56% increased likelihood of PTC diagnosis for each doubling of its concentration [9]. However, despite these findings, the literature remains equivocal regarding the association between PFAS exposure and PTC risk [17,18]. Because these studies differ significantly in design, number of patients, and analysis methods, it is difficult to reach a consensus on the true effects of PFAS on PTC risk. Further research is needed to better categorize the risk of environmental PFAS exposure in relation to PTC etiology and address the undue burden of disease mortality shouldered by Black and Hispanics, who also experience greater exposure to harmful levels of PFAS in drinking water [19].

In 2023, the U.S. Environmental Protection Agency (EPA) released the PFAS Analytic Tools, a web-based data repository providing accessible resources to inform the public on all locations in the U.S. where PFAS manufacturing, release, and exposure may cause adverse health effects [20]. These tools represent the EPA’s efforts to advance understanding of health risks associated with environmental PFAS exposure. Leveraging the PFAS Analytic Tool’s capacity, this study investigated (1) the relationship between socioeconomic conditions and environmental PFAS exposure as measured by proximity to PFAS sites and public drinking water PFAS testing status and (2) the relationship between PTC incidence and environmental PFAS exposure in Florida.

## 2. Materials and Methods

### 2.1. Data Sources

#### 2.1.1. Cancer Incidence and Demographic Data

Cancer incidence and socioeconomic data for this study were sourced from SCAN360 [21], a web-based data repository hosted by the Sylvester Comprehensive Cancer Center. SCAN360 integrates data from Florida Cancer Data System (FCDS), the state’s official cancer registry with socioeconomic data from the annual American Community Survey (ACS). Number of PTC cases diagnosed in Florida from 2011 to 2020 were retrieved for this analysis. The residential location for each case in FCDS was geocoded to the census tract encompassing each patient address. For every populated census tract, we first calculated the average number of PTC cases per year by the three most dominant racial and ethnic groups, including non-Hispanic White (NHW, 61.6% of total Florida population in 2020), Hispanic (18.7%), and non-Hispanic Black (NHB, 12.4%). We then calculated age-adjusted PTC incidence (i.e., new diagnosed cases per 100,000 population per year) for all census tracts using the 2011 census tract population as the basis (i.e., 2010 decennial census population numbers multiplied by a growth rate estimated with 2010 and 2020 decennial census populations) with age-adjustment weights based on the year 2000 standard population from the Surveillance, Epidemiology, and End Results Program [22]. We decided not to analyze the incidence by other racial and ethnic groups due to their relatively small (6% of total population) population sizes, which compromised statistical power. For investigation of socioeconomic variability and PTC risk, we merged the incidence data with socioeconomic variables from the ACS 5-year estimates of 2015 (i.e., mid-year between 2011 and 2020) [23]. We also merged behavioral risk variables from the U.S. Center for Disease Control and Prevention (CDC) Places data including known risk factors for PTC development such as rates of obesity and lack of physical activity.

#### 2.1.2. Environmental PFAS Exposure Data

Data downloaded from EPA’s PFAS Analytical Tool in 2024 were used to identify the locations (i.e., by geographic longitudes and latitudes) of Florida sites that can potentially release PFAS into the environment, including federal (e.g., military bases) and industrial facilities handling (i.e., manufacturing, shipping, and/or using) PFAS containing products, National Priorities List Superfund sites with PFAS detections, and sites where spills of PFAS products (e.g., aqueous film-forming foam used to combat flammable liquid fires) occurred in the past. Cleaning of the PFAS location data was then undertaken to remove duplicate listings. We also identified from the PFAS Analytical Tool ZIP code areas in Florida that were served by public water systems (PWS) testing positive for PFAS (i.e., the concentration of PFAS in the water was above EPA’s reporting thresholds) [20].

### 2.2. Data Availability

Except for cancer registry data from FCDS, all other data used for this study were derived from public sources (i.e., U.S. Census and EPA) and are available upon request from the corresponding author. Cancer related data were available after obtaining authorization from FCDS.

#### Measurement of Proximity to PFAS Sites

Calculation of a census tract’s proximity to PFAS sites followed the methodology that the EPA developed for measuring proximity to sites of environmental concerns [24]. ArcGIS Pro 3.0 was used for the calculation. Because a census tract is made up of census blocks (i.e., the smallest census population unit) with varying number of populations, proximity index for a census tract was calculated as a population-weighted average proximity of all blocks within the census tract (see Equation (1)). Weighing a census block’s proximity measurement by its population ensures that zero-population blocks do not factor into the calculation, and exposure risks experienced at blocks with large populations are proportionally reflected in the index.(1)$$P_{k}=\frac{\sum_{i,j}p_{i}*\frac{1}{d_{i,j}}}{\sum_{i}p_{i}}$$
where *P_k_* = PFAS proximity index for census tract *k*

*i* = a census block within the census tract

*j* = a PFAS site within 5 km from the centroid of block *i*

*d_i_*_,*j*_ = distance (km) between the centroid of block *i* and PFAS site *j*

*p_i_* = population of census block *i*

On rare occasions, calculation of the proximity index for a particular census tract was zero because there may not have been any sites located within 5 km of every block within the census tract. For these census tracts, the proximity index was calculated with a radius of 10 km (or 15 km if measurement by 10 km was also zero).

### 2.3. Statistical Analysis

Associations between environmental PFAS exposure (i.e., proximity to PFAS sites and PWS with positive PFAS status) and age-adjusted incidence of PTC at the census tract level were analyzed with contingency tables and stepwise regression analysis. All variables analyzed are included in Table 1. For contingency table analysis, a categorical variable denoting 3 levels of PFAS proximity was created by first ranking census tract PFAS proximity indices from highest to lowest, then dividing all census tracts into three equal-sized groups based on the ranking. The highest one third was assigned level 3 and the lowest one third was assigned level 1. Two-way ANOVA was used to compare mean incidence and socioeconomic disparities for the three levels of PFAS proximity, while two-sample t-test was used to compare means for census tract drinking water PFAS status (i.e., served by PWS tested positive for PFAS above reporting threshold versus PWS tested negative for PFAS above reporting threshold). ArcGIS Pro 3.0 by ESRI (Redlands, CA, USA) was used to perform geospatial analysis. This article is a revised and expanded version of an abstract entitled “Abstract 2309: An ecological study on environmental exposure to PFAS and thyroid cancer incidence by races/ethnicities in Florida,” which was presented at the American Association for Cancer Research Annual Meeting 2025, Chicago, IL, USA, 25–30 April 2025 [25].

## 3. Results

A total of 14,939 PTC cases were diagnosed in Florida between 2011 and 2020 with NHW participants accounting for 64% (*n* = 9169) of the total cases and females accounting for 74% (*n* = 10,496). The average age at diagnosis was 50.6 (SD = 16.2) years. There was a total of 880 deaths (6.2%) during the study period. Table 1 shows statistics of PTC patients in Florida from 2011 to 2020, including case summaries, demographic characteristics, the average values (i.e., by census tracts of patients’ residential addresses) of socioeconomic variables, behavioral risk factors for PTC, average number of PFAS sites within 5 km from the centroid of a census tract, and the percentages of census tracts served by PFAS positive PWS. Looking specifically at socioeconomic data, NHB and Hispanic patients had a lower median household income and a lower percentage of individuals with a bachelor’s degree compared to the NHW population. NHB and Hispanic individuals more frequently lived in poverty and had no health insurance. NHB and Hispanic populations also had a higher average number (13) of PFAS facilities within 5 km of residential address compared to NHW populations (9).

**Table 1 ijerph-22-01290-t001:** Descriptive Statistics of Papillary Thyroid Cancer Patients (*n* = 14,414) by Races/Ethnicities and Socioeconomic/Environmental Factors by Patients’ Residential Census Tracts in Florida from 2011 to 2020.

Demographics	NHW	Hispanic	NHB	Total
2015 Florida Population	11,503,781	4,599,699	2,942,326	19,045,806
Number of Patients	9169	3990	985	14,144
Percent Female	71.6%	78.4%	81.7%	74.2%
Average Age at Diagnosis	52.46	48.86	50.56	50.63
Number of Mortality	647	147	86	880
Mortality Odds ^1^	0.07	0.04	0.09	0.06
Socioeconomic Measures				
Average Median Household Income	57,569	52,139	46,010	51,906
Average Percentage of Population with a Bachelor’s Degree	30.94	27.43	22.11	26.83
Average Percentage of Population in Poverty	13.10	17.43	20.82	17.12
Average Percentage of Population with No Insurance	14.73	22.05	20.37	19.05
Average Percentage of Population Lacking Physical Activity	25.97	30.72	31.54	29.41
Average Percentage of Obesity in Population	28.80	29.95	33.57	30.77
PFAS Contamination				
Average Number of PFAS Facilities within 5K	18	13	13	11
Percent Served by PWS with Positive PFAS	0.29	0.30	0.36	0.31

^1^ Number of mortality/number of patients.

Figure 1 shows the geographic distribution of PFAS sites, zip code areas with positive PFAS status, and the number of populations by counties in Florida. The statewide map shows that zip code areas served by PWS with PFAS status tend to concentrate in areas with high numbers of PFAS sites. Six counties were identified as having the highest numbers of PFAS sites, including Duval (containing Jacksonville), Hillsborough (Tampa), Broward (a part of the Miami metropolitan area), Pinellas (a part of the Tampa metropolitan area), Miami-Dade (city of Miami), and Orange (Orlando). Together, these six counties represented 44% of Florida’s total population in 2023 (24), demonstrating how environmental exposure to PFAS has affected a large portion of Florida’s population.

Figure 2 maps the locations of PFAS facilities, ZIP code areas with positive PFAS testing, and income levels by census tracts in the four metropolitan areas labeled (i.e., Miami, Tampa, Orlando, and Jacksonville). We found that there are more PFAS sites in lower income census tracts than those with higher incomes, demonstrating socioeconomic disparities in environmental exposure to PFAS contamination.

Table 2 shows the contingency table with race and ethnicity-specific mean values of PTC rates, socioeconomic status, and PWS PFAS status by level of proximity to PFAS sites. Two-way ANOVA confirms that proximity to PFAS sites is significantly associated with worse socioeconomic conditions as well as higher percentages of positive PWS PFAS status. For NHW and Hispanic populations, age-adjusted incidence of PTC increase as PFAS proximity level increases; however, no such association was found between PTC incidence and PFAS proximity in NHB patients. in NHW populations.

Table 3 shows statistically significant associations between census tract socioeco-nomic disparities and positive PFAS status in drinking water. Census tracts served by PWS with positive PFAS status also show higher PTC incidence in all three groups; how-ever, this association is only statistically significant at a 90% confidence level (*p* = 0.07).

In Table 3, it can be seen that higher PFAS proximity is associated with higher obesity rates (i.e., a known risk factor for PTC) in the population. To assess if obesity was a confounder in the observed association between PFAS exposure and PTC incidence, we retrieved weight and height data from FCDS to calculate patients’ body mass index (BMIs) (Table 4). Because weight and height data are not mandated items of FCDS, of the 14,144 cases diagnosed, only 4918 cases (35%) contained valid weight and height data for BMI calculation (i.e., 65% missing rate). Table 4 shows no statistically significant associations between census tract PFAS proximity level and average BMI in patients with valid data, mostly because the differences in average BMIs by PFAS proximity levels are very small.

To assess if the observed associations between PTC rates and PFAS exposure were mediated by confounding variables, we performed stepwise regression analyses with census tract age-adjusted incidence of PTC as dependent variables and all socioeconomic, behavioral risk, and PFAS exposure variables from Table 1 and Table 2 as independent variables (Table 5). Census tract PFAS proximity index was analyzed as a continuous variable in the regression models. The proximity index for PFAS sites within 5 km remains a significant variable in both the NHW and Hispanic models. The percentage of the population with no insurance is the other significant variable in the NHW model. PWS PFAS status (i.e., a dummy variable) by itself is not significant in any of the three models. For the NHB model, proximity to PFAS sites measured with a 5 km radius is not significant. When the proximity index was recalculated with a search radius of 20 km, proximity to PFAS site was the only significant factor for NHB PTC incidence.

## 4. Discussion

By merging data from EPA’s PFAS Analytic Tool with SCAN360, we identified associations between socioeconomic disparities and greater levels of environmental PFAS exposure in census tracts disproportionately occupied by lower-income Hispanic and NHB populations in Florida. We also found that census tracts with higher degrees of proximity to PFAS sites were more likely to be served by PWS that tested positive for PFAS, suggesting that concentration of PFAS sites near PWS may lead to drinking water contamination with PFAS. This is consistent with previous statewide studies that have demonstrated widespread PFAS detection in drinking water across all 67 counties in Florida with higher concentrations near potential point sources located in or near metropolitan areas, including airports, military bases, and wastewater facilities [26,27,28]. For example, previous studies have found that the number of PFAS sites within a community water system’s watershed is positively associated with the frequency and concentration of PFAS concentration detected in drinking water [19], and the presence of industrial sites, military fire training areas, and wastewater treatment plants predict PFAS detection frequency and concentrations in water systems [29]. This is concerning because sites that can potentially release PFAS into the environment are mostly located in highly populated metropolitan areas, and uncontrolled release of PFAS chemicals can impact the health of a large portion of Florida’s population with a greater burden on minoritized populations [19]. In addition, our study identified associations between proximity to PFAS sites and age-adjusted PTC incidence in NHW and Hispanic populations. With patient weight and height data available for this analysis, we did not find evidence supporting obesity as a confounding variable mediating the observed associations between PFAS proximity and PTC incidence. It is important to note that while serum PFAS levels have been linked to altered levels of thyroid hormone, they have not been consistently associated with BMI, suggesting that PFAS mediated changes in BMI is unlikely to confound our observed associations between PFAS exposure and PTC risk [30].

Limitations of this study are related to availability as well as the timeliness of the data. For example, the metropolitan areas of Jacksonville and Orlando contain a high number of PFAS sites, but most ZIP code areas located within the two metro areas do not show positive PWS PFAS status. EPA mandated that all PWS in the U.S. complete initial monitoring for PFAS in its water by 2027, and we downloaded and analyzed the data from EPA in October 2024. We do not know if PWS serving the Jacksonville and Orlando metro areas have completed their analysis by the time we downloaded the data. When all PWS in Florida have completed their PFAS testing by 2027, future studies can be conducted to assess the associations between exposure to different PFAS in drinking water and incidence of cancer types whose risks are theoretically linked to these chemicals. In addition, PFAS exposure was assessed ecologically in this study using proximity to PFAS sites and PWS contamination at the census tract and ZIP code levels. While this approach captures population-level trends, it may not reflect individual-level PFAS exposure, thus limiting the granularity of exposure-outcome inferences. Further, proximity to PFAS sites does not necessarily equate to exposure, as actual contamination depends on environmental transport mechanisms, site history, and individual behaviors such as water source usage [31]. However, it is known that minoritized patients with fewer socioeconomic resources are less likely to participate in clinical trials [32], and as we have found with data from Florida, minority populations are exposed to higher levels of environmental PFAS contamination. It is possible that existing research on PFAS and cancer risk that relied on clinical trial data may not contain sufficient samples from these disadvantaged populations facing high levels of environmental PFAS exposure. Strengths of our study include the combination of the EPA’s PFAS Analytic Tool with FCDS through SCAN360, as well as ACS and CDC PLACES data. This multi-source integration allows for a comprehensive, population-level view of PFAS exposure, cancer incidence, and community vulnerability. Further, this data triangulation increases the validity of findings and provides a replicable model for other states.

In November 2024, EPA released a report highlighting key EPA accomplishments under EPA’s PFAS strategic roadmap (established in 2021), the national strategy to protect communities across the country form PFAS contamination [33]. These accomplishments were centered around three overarching goals of the roadmap: restrict, remediate, and research. In April 2024, the EPA established the first federal, legally enforceable drinking water standards for several PFAS congeners individually and in mixtures. The EPA has also catalyzed the cleanup of PFAS contamination by designating PFOS and PFOA as hazardous substances in 2024. While these achievements will begin restricting and remediating PFAS in the environment soon, more research efforts on the associations between PFAS exposure and risk for specific health conditions are urgently needed for the development of clinical and public health interventions. Future studies should strive to sample and perform blood and/or tissue analysis based on stratifications of environmental PFAS exposure levels as demonstrated in this study to determine how PFAS exposure correlates with biological levels of thyroid stimulating hormone, cholesterols, and intra-tumoral PFAS levels. Environmental sampling is also needed to establish PFAS levels in drinking water, soil, and air for correlation analyses with various measures of PTC incidence and clinical outcomes. The results of this study highlight the urgent need for multidisciplinary collaboration between environmental science and cancer research to jointly accelerate research efforts focused on better understanding of cancer risks associated with PFAS chemical exposures.

## 5. Conclusions

This study evaluated the association between environmental exposure to PFAS and risk of PTC using a statewide database of cancer outcomes. Our results demonstrate significant environmental exposure disparities based on socioeconomic factors, as well as positive associations between residential PFAS exposure and PTC risk among NHW and Hispanic populations throughout Florida.

## Figures and Tables

**Figure 1 ijerph-22-01290-f001:**
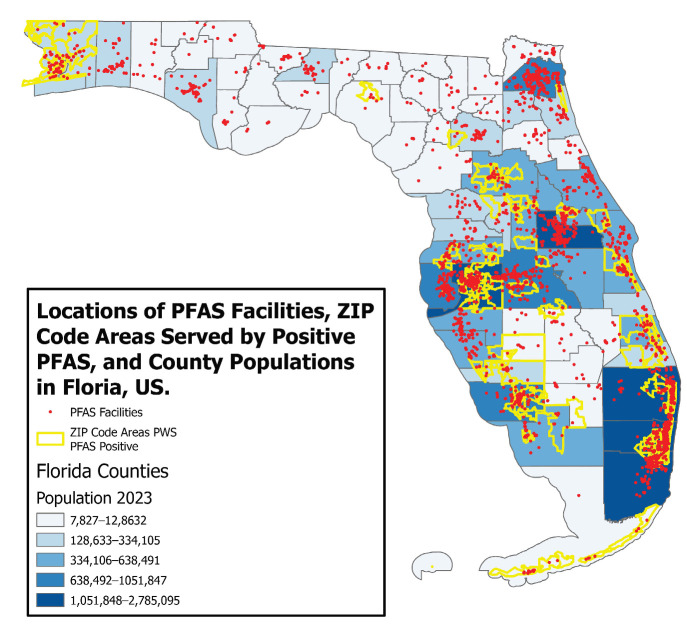
Zip code areas served by PWS with positive PFAS status tend to concentrate in areas with high numbers of PFAS sites and denser populations in Florida.

**Figure 2 ijerph-22-01290-f002:**
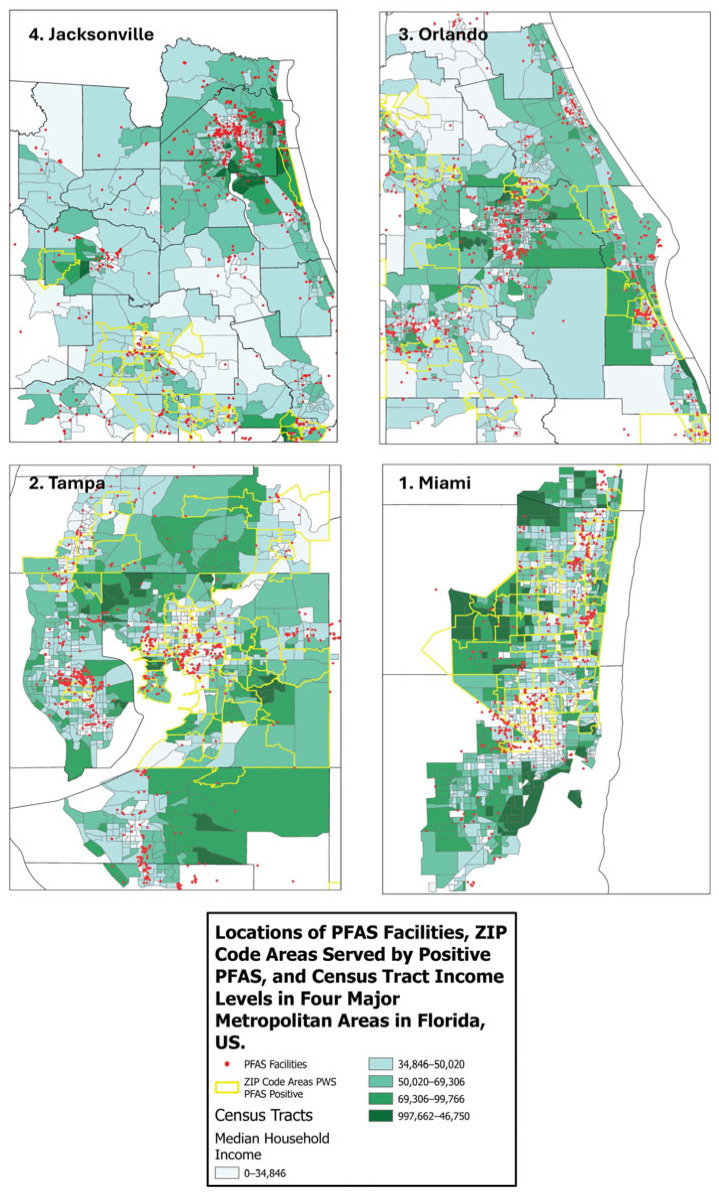
Mapping of locations of PFAS Facilities, ZIP code areas served by positive PFAS contamination, and census tract income levels in four major metropolitan areas in Florida, USA demonstrates that lower income census tracts contain more PFAS sites than those with higher incomes indicating socioeconomic disparities in terms of environmental exposure to PFAS contamination.

**Table 2 ijerph-22-01290-t002:** Papillary Thyroid Cancer Incidence and Socioeconomic Disparities by Environmental PFAS Exposure Levels per Patients’ Residential Census Tracts (*n* = 3688).

		Average Values for NHW				Average Values for Hispanic				Average Values for NHB				Average Values for All Races/Ethnicities						
Level of Proximity to PFAS Facilities ^1^	Number of Census Tract	Population	Number of Cases	Age	AAInc ^2^	Population	Number of Cases	Age	AAInc	Population	Number of Cases	Age	AAInc	Median Household Income	% in Poverty	% Without Insurance	% with Bachelor’s Degree	% Obesity	% Lack of Physical Activity	Census Tracts Serviced by PWS with Positive PFAS
1	1223	3424	3.1	53.0	0.98	1037	1.0	48.3	0.58	434	0.2	49.2	0.35	57,043	13.5	15.1	28.4	29.4	27.1	25%
2	1223	2817	2.5	53.4	0.99	1026	1.0	47.1	0.68	678	0.3	50.7	0.38	53,278	15.3	16.5	29.2	29.3	27.7	30%
3	1222	2071	1.8	51.7	1.24	1257	1.2	48.6	0.75	959	0.3	50.3	0.36	44,217	20.3	21.2	25.2	31.3	30.1	39%
Total	3668	2771	2.46	52.4	1.07	1107	1.1	48.0	0.67	690	0.3	50.2	0.36	51,511	16.4	17.6	27.6	30.0	28.3	31%
One-Way ANOVA (*p*-value)				0.73	0.006				0.005				0.78	<0.001	<0.001	<0.001	<0.001	<0.001	<0.001	<0.001

^1^ Index of proximity to PFAS Sites within 5 km of a Census Tract. Proximity levels: 3 > 2 > 1 (highest to lowest proximity indices to PFAS sites within 5 km), ^2^ Average age-adjusted incidence per census tract.

**Table 3 ijerph-22-01290-t003:** Papillary Thyroid Cancer Incidence and Socioeconomic Disparities by PWS PFAS Contamination (*n* = 3688).

		Average Values for NHW				Average Values for Hispanic					Average Values for NHB					Average Values for All Races/Ethnicities						
PWS PFAS Status ^1^	Number of Census Tract	Population	Number of Cases	Age	AAInc ^2^	Population	Number of Cases	Age		AAInc	Population	Number of Cases	Age		AAInc	Median Household Income	% in Poverty	% Without Insurance	% with Bachelor’s Degree	% Obesity	% Lack of Physical Activity	Average Proximity to PFAS Sites Within 5 km
0	2527	2924	2.5	52.3	1.03	1119	1.1	48.3	0.65	647		0.3	50.1	0.35	52,343		16.0	17.1	28.3	30.1	28.1	0.003
1	1141	2431	2.3	52.8	1.16	1080	1.0	47.4	0.71	787		0.3	50.4	0.39	49,670		17.1	18.6	26.2	29.7	28.8	0.005
One-Side *t*-Test Significance (*p*-value)			0.07						0.12					0.20	<0.001		0.003	<0.001	<0.001	0.009	0.003	<0.001

^1^ PWS PFAS status (0 = Served by Public Water Systems whose PFAS Levels below EPA Reporting Thresholds; 1 = PFAS above EPA Reporting Thresholds), ^2^ Average age-adjusted incidence per census tract.

**Table 4 ijerph-22-01290-t004:** Average Patient BMIs (*n* = 4918) by PFAS Proximity Levels per Patients’ Residential Census Tracts.

	NHW		Hispanic		NHB		Total	
Level of Proximity to PFAS Facilities	Number of Cases	Average BMI	Number of Cases	Average BMI	Number of Cases	Average BMI	Number of Cases	Average BMI
1	1190	30.16	423	29.69	74	32.50	1807	29.91
2	966	29.80	446	29.24	98	33.25	1616	29.66
3	708	29.73	577	29.73	133	33.43	1495	29.93
Total Cases (with BMI Data)	2864	29.93	1446	2956	305	33.15	4918	29.83
One-Way ANOVA Significance (*p*-value)		0.37		0.42		0.74		0.50
PWS PFAS Status ^1^								
0	2030	29.94	1003	29.35	187	33.18	3441	29.74
1	834	29.92	443	30.04	118	33.09	1477	30.06
Total Cases (with BMI Data)	2864	29.93	1446	29.56	305	33.15	4918	29.83
One-Way ANOVA Significance (*p*-value)		0.48		0.03		0.46		0.11
Available BMI Data								
Number of Cases Missing BMI Data (% Missing)	6332 (69%)		2544 (64%)		680 (69%)		9226 (65%)	
Total Papillary Thyroid Cancer Cases	9196		3990		985		14,144	

^1^ PWS PFAS status (0 = Served by Public Water Systems whose PFAS Levels below EPA Reporting Thresholds; 1 = PFAS above EPA Reporting Thresholds).

**Table 5 ijerph-22-01290-t005:** Regression analysis of socioeconomic variables and environmental PFAS exposure associated with papillary thyroid cancer age-adjusted incidence by census tracts.

	Independent Variables				
Dependent Variable	Constant	Proximity Index for PFAS Sites Within 5 km	Percentage of Population with No Insurance	Proximity Index for PFAS Sites Within 20 km	Regression ANOVA
NHW Age-Adjusted Incidence	Beta = 0.78	Beta = 28.24	Beta = 0.01		df = 2
	Std. Error = 0.08	Std. Error = 7.24	Std. Error = 0		F = 13.847
	t = 9.66	t = 3.9	t = 2.39		Sig. < 0.001
	Sig. < 0.001	Sig. < 0.001	Sig. = 0.017		
Hispanic Age-Adjusted Incidence	Beta = 0.615	Beta = 12.744			df = 1
	Std. Error = 0.029	Std. Error = 4.363			F = 8.534
	t = 21.247	t = 2.921			Sig. = 0.004
	Sig. < 0.001	Sig. = 0.004			
NHB Age-Adjusted Incidence	Beta = 0.307			Beta = 4.392	df = 1
	Std. Error = 0.035			Std. Error = 2.097	F = 3.338
	t = 8.834			t = 2.095	Sig. = 0.036
	Sig. < 0.001			Sig. = 0.036	

## Data Availability

Additional data are available from the corresponding author on reasonable request.

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
