# Peer review of "Socioeconomic Disparities and Risk of Papillary Thyroid Cancer Associated with Environmental Exposure to Per- and Polyfluoroalkyl Substances (PFAS) in Florida"

_ijerph, 2025, doi:10.3390/ijerph22081290_

Round 1

Reviewer 1 Report

Comments and Suggestions for Authors

This study investigates the association between environmental PFAS exposure and papillary thyroid cancer (PTC) risk in Florida, with a focus on socioeconomic and racial/ethnic disparities. By integrating data from the EPA’s PFAS Analytic Tool, the Florida Cancer Data System, and socioeconomic surveys, the authors provide a comprehensive ecological analysis. The findings suggest that higher PFAS exposure is linked to increased PTC incidence in Non-Hispanic White and Hispanic populations, as well as greater burdens in lower-income communities.

The study’s strengths lie in its timely focus on PFAS, a growing public health concern, and its robust methodology, which includes geospatial analysis and population-weighted exposure indices. The emphasis on environmental justice adds significant value, highlighting disparities in exposure and health outcomes. However, the ecological design limits causal inference, as proximity to PFAS sites does not confirm individual exposure. Additionally, incomplete PFAS testing data and missing BMI values (65% missing) weaken some conclusions.

Future research could benefit from incorporating biomarker data to directly measure PFAS exposure and expanding the geographic scope beyond Florida. Despite these limitations, the study offers important insights into the intersection of environmental contamination, cancer risk, and health inequities, underscoring the need for targeted policies and further investigation.

Author Response

This study investigates the association between environmental PFAS exposure and papillary thyroid cancer (PTC) risk in Florida, with a focus on socioeconomic and racial/ethnic disparities. By integrating data from the EPA’s PFAS Analytic Tool, the Florida Cancer Data System, and socioeconomic surveys, the authors provide a comprehensive ecological analysis. The findings suggest that higher PFAS exposure is linked to increased PTC incidence in Non-Hispanic White and Hispanic populations, as well as greater burdens in lower-income communities.

The study’s strengths lie in its timely focus on PFAS, a growing public health concern, and its robust methodology, which includes geospatial analysis and population-weighted exposure indices. The emphasis on environmental justice adds significant value, highlighting disparities in exposure and health outcomes. However, the ecological design limits causal inference, as proximity to PFAS sites does not confirm individual exposure. Additionally, incomplete PFAS testing data and missing BMI values (65% missing) weaken some conclusions.

Future research could benefit from incorporating biomarker data to directly measure PFAS exposure and expanding the geographic scope beyond Florida. Despite these limitations, the study offers important insights into the intersection of environmental contamination, cancer risk, and health inequities, underscoring the need for targeted policies and further investigation.

We thank the reviewer for their comments. We have emphasized in our discussion the important limitations and next steps for future studies evaluating the health risks of exposure to PFAS that were brought up by the reviewer. We have also made edits to the introduction, methods, results, discussion, and figures as recommended by the reviewer in the checklist.

Reviewer 2 Report

Comments and Suggestions for Authors

Please align your Abstract with the Title. See my comments in the document.

You did not backup your "Discussion" section, which is a fundamental aspect in research/scientific paper. Please address it.

Please see all my comments in the manuscript.

Author Response

Please align your Abstract with the Title. See my comments in the document. You did not backup your "Discussion" section, which is a fundamental aspect in research/scientific paper. Please address it. Please see all my comments in the manuscript.

We thank the reviewer for their thoughtful comments and feedback throughout our manuscript. We have adjusted the title and abstract to be more aligned with one another. We also edited the discussion to support our results with existing literature. Lastly, we responded to all the comments left in the manuscript. Please see below for our response to each specific comment.

  1. The Title of your article suggests otherwise. It says "Socioeconomic disparities associated with", which means the Social aspects (e.g., education, family structure, settlement, etc.) of the residents in Florida led them to be exposed to these substances, the same with their economic (e.g., employment status, job type, etc.) situation. So this is the focus of your study. The title does not say "Association between Environmental Exposure to Per- and Polyfluoroalkayl Substances and Papillary Thyroid cancer risks".

We agree with the reviewer that the original structure of our abstract did not discuss our findings of the relationship between socioeconomic status and environmental PFAS exposure. We have adjusted the abstract to include these findings. We also updated the title to clarify the associations that we evaluated in our study. (Page 1, Lines 2-3 and 17-33)

  1. You have not mentioned anything about "socioeconomic disparities" in this conclusion. Does this mean you do not need to include "Socioeconomic Disparities"? You may change the title to "Association between environmental PFAS exposure levels and increased PTC risks". The "socioeconomic" aspect is inherently integrated into your methodology, results and discussion.

We thank the reviewer for this important comment. We agree it is important to highlight the disparities that we found in our analysis that are critical to our paper in the abstract section. We have adjusted our abstract to ensure that all major findings in our study are included in the abstract. (Page 1, Lines 17-33)

  1. What about "Socioeconomic disparities"? If you have reached the maximum number of keywords, you may replace "environmental justice" with it. Please write it in full [PFAS]. You may write "Papillary Thyroid Cancer".

We agree that “Socioeconomic Disparities” is an important keyword and have included it in our list. We also changed “Thyroid Cancer” to “Papillary Thyroid Cancer” to more accurately reflect the content of our manuscript. (Page 1, Lines 34-35)

  1. What do you mean by this [in the context of the US in general and your study area in particular?]. This is important due to the high prevalence of PTC and the mortality rates.

In this sentence, we discuss the disparities that exist in the United States within papillary thyroid cancer outcomes to highlight that minoritized groups have worse survival rates and higher mortality, which is often due to delays in diagnosis and treatment. This sentence is specifically discussing outcomes within the United States in general to support our rational for analyzing socioeconomic disparities in our statewide cancer database. We have added “in the United States” to this sentence to clarify. (Page 2, Lines 42-43)

  1. What are you referring to here, "Replacement Therapy" or "costs associated with the treatment and maintenance of Thyroid Cancer"?

We agree with the reviewer that we should be more precise when discussing the associated costs of papillary thyroid cancer. We have updated this sentence to include all the components that make up the total cost to the US health system of thyroid cancer. (Page 2, Lines 46-48)

  1. Is it "healthcare or health care"? Please be consistent.

We thank the reviewer for recognizing this inconsistency. We have changed all “health care” to “healthcare” throughout the manuscript to be consistent. (Page 2, Lines 46 and 49)

  1. I suggest you consistently write Papillary Thyroid Cancer or PTC.

We agree with the reviewer and have ensured we consistently wrote PTC throughout the manuscript after defining this acronym in the second sentence of the introduction.

  1. You must incorporate this in the Abstract, where I said the title says otherwise.

We have adjusted the abstract to include information regarding our analysis and findings on the relationship between socioeconomic conditions and environmental PFAS exposure. Please see responses to comments 1 and 2 for specific changes. (Page 1, Lines 17-33)

  1. Please delete [rates].

We have deleted rate after incidence throughout the paper.

  1. Is this an Annual Survey [American community survey]?

We agree with the reviewer that we should specify the frequency that this survey is conducted, which is important to understand the data source that we used for our analysis. As this is an annual survey, we clarified this in this sentence. (Page 2, Line 79)

  1. Please do not begin a sentence with an abbreviation.

We have changed this sentence, so it no longer starts with an abbreviation. (Page 2, Line 79)

  1. incidence means "new cases". Is that what you mean here? Otherwise you should talk about "Prevalence, which is the number of existing cases at a particular time.

We thank the reviewer for this comment. As we calculated new cases diagnosed per 100,000 population per year (i.e., the incidence), we have changed the description of age-adjusted incidence to align with the proper terminology. (Page 3, Line 84)

  1. I take it that this is just a statement. Did this happen in your study? If so, you must write what you have done. I have highlighted "is" to draw your attention to the "present tense", meaning you are not referring to your study. Remember you must use past tense if you are writing about what you did.

We agree with the reviewer that the way this sentence was originally written was confusing and does not accurately convey the methods of our analysis. We have edited this sentence to reflect the procedure we used to calculate the proximity index and changed all current tense verbs to past tense. (Page 3, Lines 122-124).

  1. Please introduce Table 1 here. For example: ...population (9). Statistics of Papillary Thyroid Cancer Patients in Florida from 2011 to 2020 are presented in Table 1.

We have included this description of our introduction of Table 1 in the Results section. We elected to keep the original location of our introduction of Table 1 as we describe the important findings of Table 1 following this sentence. (Page 4, Lines 141-145)

  1. This is not clear, please clarify this. Please give context. (line 151)

We thank the reviewer for noticing this error and have removed the content that does not align with the table description. We have updated the descriptions for Table 1 superscripts accordingly. (Page 5, Line 154)

  1. Figure caption must be at the bottom of the Figure. Only a Table caption must be at the top of the Table. Again, when you write anything about a Figure/Table, you must first introduce the Figure/Table, then present it and then briefly describe it. Please follow this format with your Figures/Tables.

We thank the reviewer for this comment. We have edited each table caption to ensure the caption is above the tables and the superscript descriptions below the tables are updated. In addition, we have updated Figure 1 and created a Figure 2 to improve the readability of these maps. Finally, we removed the previous title that was above Figure 1 and updated the captions that are now below each figure.

  1. This part does not say anything about "results" but more of a methodology, which you have already mentioned in the methodology section. Please remove it.

We have removed the first sentence from the Figure 1 caption that did not discuss the results of the map from the figure caption. (Page 6, Lines 183-184)

  1. Your maps are too small and difficult to analyze. Is it not possible for you to increase them?

We appreciate the reviewers comment regarding the readability of our maps. We have increased the map sizes as well as made two separate figures to more easily read the map inserts.

  1. Which map are you referring to? What is its number?

This map was referring to the large map of Florida from our original submission. We have now separated the statewide map (Figure 1) from the map of 4 major metropolitan areas in Florida (now Figure 2). We believe this improves the readability of our maps. We have also edited the figure legends of both Figure 1 and Figure 2 to highlight the major findings of the maps/figures.

  1. You did not introduce this Table. Consider my comments in line 167, which is: When you write anything about a Figure/Table, you must first introduce that Figure/Table, then present it and then briefly describe it. Please follow this format with all your Figures/Tables.

We agree with the reviewer that is imperative to introduce all our tables and figures in our Results section text. We have ensured that we have introduced all our tables and figures more clearly in the text preceding each.

  1. Is there any reason why you did not use any literature to back up your results? You must support your results/findings with literature.

We thank the reviewer for this comment and agree our discussion should include support from the existing literature. We have updated our discussion to include information from previous studies that support our findings. We believe this has strengthened the interpretation of our results and supports and contributes to the previous literature related to the topics of environmental disparities and health effects of PFAS exposure to the population. (Pages 9-11)

Round 2

Reviewer 1 Report

Comments and Suggestions for Authors

The authors have addressed my concerns and I don't have further comments.